# Sitravatinib in combination with nivolumab plus ipilimumab in patients with advanced clear cell renal cell carcinoma: a phase 1 trial

Pavlos Msaouel [1,2,3,10] ✉, Kai Yu[4], Ying Yuan[5], Jianfeng Chen[1,3], Xinmiao Yan[4], Menuka Karki[1,3], Fei Duan[1,3], Rahul A. Sheth [6], Priya Rao[7], Kanishka Sircar[2,7], Amishi Y. Shah [1], Amado J. Zurita [1], Giannicola Genovese [1,3,4], Min Li[1,3], Chih-Chen Yeh[1,3], Minghao Dang [4], Guangchun Han [4], Yanshuo Chu[4], Max Hallin[8], Peter Olson[8], Rui Yang[8], Daniela Slavin[8], Hirak Der-Torossian [8], Curtis D. Chin[8], Nizar M. Tannir[1], Linghua Wang [4,9,10] ✉ & Jianjun Gao [1,3,10] ✉

We conducted a phase I trial to determine the optimal dose of triplet therapy with the tyrosine kinase inhibitor sitravatinib plus nivolumab plus ipilimumab in 22 previously untreated patients with advanced clear cell renal cell carcinoma. The primary endpoint was safety. Secondary endpoints were objective response rate (ORR), disease control rate (DCR), duration of response (DOR), progression-free survival (PFS), overall survival (OS), 1-year survival probability, and sitravatinib pharmacokinetics. Sitravatinib dose of 35 mg daily plus nivolumab 3 mg/kg and ipilimumab 1 mg/kg resulted in high frequency of immune-related adverse events. Subsequent dose reduction of ipilimumab to 0.7 mg/kg allowed safe escalation of sitravatinib up to 100 mg daily. Overall, the triplet combination achieved ORR 45.5%, DCR 86.4%, median PFS 14.5 months, and 1-year survival 80.8%. Median OS and DOR were not reached. Sitravatinib exposure increased dose-dependently. Single-cell RNA-seq of longitudinally collected tumor biopsies from 12 patients identified a tumor cell-specific epithelial-mesenchymal transition-like program associated with treatment resistance and poor outcomes. Treatment resistance was characterized by a transition from cytotoxic to exhausted T cell state and enrichment for M2-like myeloid cells. The observed hypothesis-generating changes in gene expression dynamics and cellular states may help inform future strategies to optimize immunotherapy efficacy. Clinical Trials.gov identifier: NCT04518046

Immune checkpoint therapy (ICT) offers durable and, in some cases, curative responses in subsets of patients with advanced clear cell renal cell carcinoma (ccRCC)[1]. Dual ICT combining the programmed death-1 (PD-1) inhibitor nivolumab with the cytotoxic T-lymphocyte associated protein 4 (CTLA-4) inhibitor ipilimumab can produce durable complete responses in approximately 10% of systemic treatment-naïve patients

with advanced ccRCC[2]. Nonetheless, a significant number of patients exhibit primary or acquired resistance to ICT[3]. Therefore, novel ICT strategies are needed to overcome primary resistance, prevent acquired resistance and improve outcomes in a wider range of patients.

Sitravatinib is a tyrosine kinase inhibitor (TKI) that targets the TYRO3, AXL, and MERTK (TAM) receptors, the vascular endothelial

**Fig. 1 | Trial design and conduct. a** Study schema. Tumor biopsy samples were collected at baseline, prior to the second infusion of the triplet therapy (approximately 3 weeks after treatment initiation), and at disease progression. **b** CONSORT diagram. **c** Treatment course timeline for the seven patients enrolled in cohort 1.

C2, cycle #2; ccRCC, clear cell renal cell carcinoma; CR complete response, DLT dose-limiting toxicity, EOT end of treatment, PD progressive disease, PR partial response, SD stable disease, Q3W every 3 weeks, Q4W every 4 weeks.

**Table 1 | Patient baseline demographics and clinical characteristics**

| Age, years | Median (IQR) | 58 (51–62) |
|---|---|---|
| | Range | 36–76 |
| **Sex, *n* (%)** | Male | 17 (77.3) |
| | Female | 5 (22.7) |
| **Ethnicity, *n* (%)** | White | 17 (77.3) |
| | Black | 2 (9.1) |
| | East Asian | 2 (9.1) |
| | South Asian | 1 (4.5) |
| **IMDC prognostic risk group, *n* (%)** | Favorable | 1 (4.5) |
| | Intermediate | 19 (86.4) |
| | Poor | 2 (9.1) |
| **Baseline peripheral blood NLR** | Median (IQR) | 2.38 (1.85–3.74) |
| | Range | 1.07–7.29 |
| **Karnofsky performance status score, *n* (%)** | 100 | 14 (63.6) |
| | 90 | 1 (4.5) |
| | 80 | 5 (22.7) |
| | 70 | 2 (9.1) |
| **Previous nephrectomy, *n* (%)** | Yes | 15 (88.2) |
| | No | 7 (11.8) |
| **Previous radiotherapy, *n* (%)** | Yes | 2 (9.1) |
| | No | 20 (90.9) |
| **Target lesions per RECIST 1.1, *n* (%)** | 1 | 6 (27.3) |
| | ≥2 | 16 (72.7) |
| **Most common active sites of metastasis, *n* (%)** | Lung | 19 (86.4) |
| | Lymph node | 14 (63.6) |
| | Bone | 4 (18.2) |
| | Liver | 2 (9.1) |
| **Sarcomatoid/rhabdoid histology, *n* (%)** | Yes | 6 (27.3) |
| | No | 16 (72.7) |

*IMDC* International Metastatic RCC Database Consortium, *IQR* interquartile range, *NLR* neutrophil-to-lymphocyte ratio, *RECIST* Response Evaluation Criteria in Solid Tumors

growth factor receptor (VEGFR) family, c-Kit, and c-MET[4]. A phase 1/1b study showed that sitravatinib monotherapy can produce antitumor responses in heavily pre-treated patients with advanced ccRCC[5]. A subsequent phase 1-2 trial found that the dual combination of sitravatinib with nivolumab in patients with advanced ccRCC refractory to prior antiangiogenic therapy yielded high clinical activity with objective response rate (ORR) of 35.7%, manageable toxicity, and promising correlative evidence that sitravatinib can produce a more favorable tumor immune microenvironment (TME)[6]. Additional tissue correlatives from a neoadjuvant phase 2 trial of sitravatinib plus nivolumab in patients with ccRCC demonstrated upregulation of the interferon gamma (IFN-γ) response pathway which may additionally sensitize tumors to anti-CTLA-4 ICT[7]. Furthermore, anti-CTLA-4 therapy enhances the efficacy of PD-1 inhibition through distinct but complementary pathways to those targeted by sitravatinib. We therefore hypothesized that the triplet combination of sitravatinib with nivolumab plus ipilimumab will lead to more potent antitumor responses in treatment-naïve patients with advanced ccRCC.

In this work, we report the results of a phase 1 trial designed to determine the optimal dose of sitravatinib to combine with nivolumab plus ipilimumab, as well as assess the safety, preliminary efficacy, and immunomodulatory effects of this triple combination as first-line therapy for patients with advanced ccRCC. The addition of sitravatinib to nivolumab 3 mg/kg and ipilimumab 1 mg/kg results in high frequency of immune-related adverse events. Dose reduction of ipilimumab to 0.7 mg/kg enables safe administration of sitravatinib up to the maximum dose of 100 mg daily. Correlative single-cell RNA

sequencing on longitudinally collected tumor samples reveals that the triplet therapy induces induces dynamic shifts in the tumor microenvironment, including increased T cells and decreased myeloid and endothelial cells, with T cell abundance correlating with better outcomes and fibroblast enrichment predicting resistance. A distinct tumor cell population enriched in epithelial-mesenchymal transition (EMT), hypoxia, and glycolysis signatures is linked to poor outcomes, while T cell exhaustion and immunosuppressive myeloid phenotypes emerge as key contributors to resistance, underscoring the need for further validation and tailored therapies.

## Results

### Patients

A total of 22 patients were enrolled from September 2020 to July 2022 (Fig. 1). The first and last patients started triplet therapy on September 16th 2020 and July 21st 2022, respectively. At data cutoff of August 25th 2023, the median follow-up time was 15.7 months and 7 of the 22 patients (32%) continued to receive treatment on trial. Table 1 lists the demographic and clinical characteristics of the 22 patients enrolled.

### Safety and dose finding

The primary endpoint of the trial was safety. Given the potential immunogenicity of the triplet therapy that can result in delayed immune-related adverse events (irAEs), the trial protocol prespecified a long dose-limiting toxicity (DLT) evaluation window of up to 9 weeks to guide dose-escalation decisions. The first cohort of patients (Cohort 1) was treated at the starting sitravatinib dose of 35 mg daily combined with nivolumab 3 mg/kg and ipilimumab 1 mg/kg (Fig. 1b). The first patient enrolled did not experience any DLTs or notable irAEs (ID#001 in Fig. 1c). The second patient (ID#002 in Fig. 1c) developed biopsy-confirmed grade 3 immune-related colitis after the second infusion of nivolumab plus ipilimumab, which successfully improved to grade 1 within 3 weeks with steroids followed by vedolizumab maintenance. The irAE therefore did not fulfill DLT criteria per protocol; the patient restarted all study medications after resolution and received enough doses to be DLT-evaluable. The third patient (ID#003 in Fig. 1c) experienced immune-related grade 2 myocarditis near the end of cycle 2, confirmed by cardiac biopsy, that also did not fulfill DLT criteria per protocol because it was mild and resolved to baseline well within 3 weeks with steroid treatment and plasmapheresis. Since nivolumab and ipilimumab were discontinued as per standard of care treatment guidelines for immune-related myocarditis, the patient did not receive enough doses to be considered DLT-evaluable. To ensure safety prior to dose escalating sitravatinib, a fourth patient (ID#004 in Fig. 1c) was subsequently enrolled in this cohort and developed DLT of muscle biopsy-confirmed grade 3 immune-mediated myositis / myasthenia gravis that improved with high-dose steroids, plasmapheresis, and pyridostygmine. Three more patients were therefore enrolled in this cohort per trial protocol for a total of 7 patients (Fig. 1b) to confirm safety prior to dose escalation given the known immunogenicity of sitravatinib. Although none of these last three enrolled patients in cohort 1 formally experienced a DLT within 9 weeks, the first patient (ID#005 in Fig. 1c) developed grade 3 immune-related myalgias / arthralgias after the 9-week DLT window which resolved on high dose steroids within 3 days. The patient was concurrently started on atovaquone prophylaxis while on steroid therapy and approximately 16 weeks later developed fatal acute hepatitis possibly related to trial therapy or atovaquone. Liver biopsy revealed neutrophil rich infiltrates not typically observed in ICT hepatotoxicity. The next patient (ID#007 in Fig. 1c) experienced biopsy-confirmed grade 3 immune-related colitis after 9 weeks successfully treated with steroids and vedolizumab. The last patient in this cohort (ID#009 in Fig. 1c) developed grade 3 immune-related pneumonitis within 5 weeks of treatment initiation which quickly resolved within 4 days with high dose steroids. One dose of infliximab was also administered which was complicated by grade 3

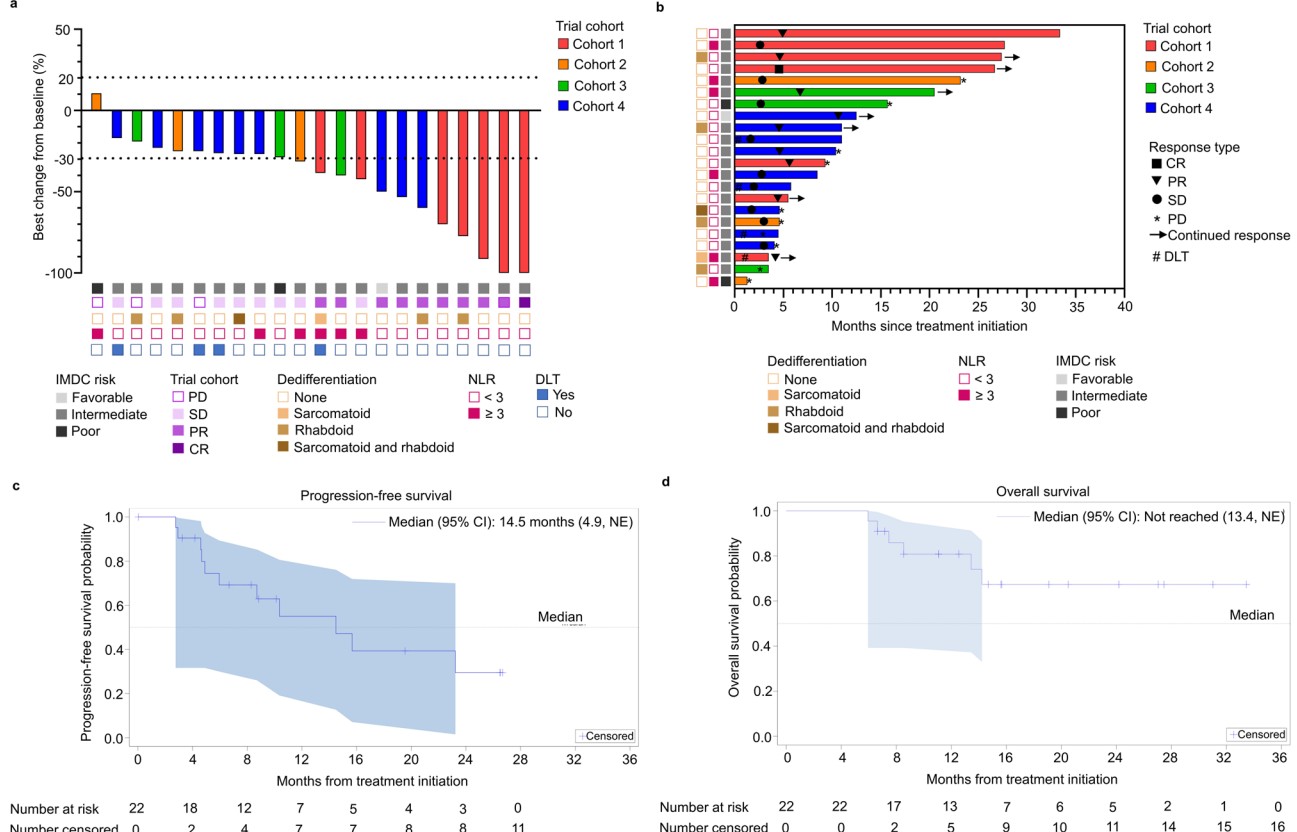

**Fig. 2 | Efficacy outcomes of sitravatinib in combination with nivolumab and ipilimumab. a** Waterfall plot showing the maximum change in the sum of the longest dimensions in each of the 22 patients treated with sitravatinib + nivolumab + ipilimumab. The IMDC risk category, presence or absence of sarcomatoid and/or rhabdoid dedifferentiation, as well as the baseline peripheral blood neutrophil-to-lymphocyte ratio (NLR) for each patient are also shown. **b** Swimmer plot showing the durability of responses to therapy. The IMDC risk category, presence or absence of sarcomatoid and/or rhabdoid dedifferentiation, and NLR for each patient are also shown. **c, d** Kaplan-Meier survival curves for progression-free survival (**c**) and overall survival (**d**) in all 22 patients treated with sitravatinib + nivolumab + ipilimumab. Dotted lines intersect the survival curve at median (also printed). Shaded blue areas show 95% Hall-Wellner confidence bands. Source data are provided in the Source Data file.

transaminitis that spontaneously resolved. Approximately one month later, after the DLT evaluation period, the patient developed grade 1 immune-related acute kidney injury that also quickly resolved within 4 days of high dose steroid initiation. The patient has continued on sitravatinib monotherapy and the metastatic tumor remains in complete response (CR) with no further irAEs noted. The ICT backbone of nivolumab plus ipilimumab was discontinued in all six patients on cohort 1 as soon as they developed irAEs (Fig. 1c) and sitravatinib monotherapy was continued until disease progression, death, or loss to follow-up.

Despite not formally counting as DLTs per protocol, the safety concerns regarding the frequency, characteristics, and quality of irAEs occurring in 6 out of 7 patients treated with the triplet therapy in cohort 1 (Fig. 1c) were strong enough to deem this regimen as an unlikely viable for further investigation. Considering the strong early signs of efficacy in this cohort with an objective response noted in 6 out of 7 patients (Fig. 1c), the trial protocol was revised so that cohort 2 would de-escalate the dose of ipilimumab to 0.7 mg/kg for up to four doses, to mitigate the development of the irAEs observed in cohort 1, based upon our suspicion that ipilimumab was the major agent responsible for the irAEs and published data suggesting that lower ipilimumab doses can maintain efficacy while decreasing toxicity[8,9]. Given the often early onset of irAEs in cohort 1 (Fig. 1c), we chose to reduce the ipilimumab dose rather than maintaining the same dose and either reduce the total lifetime number of doses to less than four or increase the administration interval to longer than every 3 weeks.

No DLTs or other concerning irAE signals were observed in the three patients enrolled in cohort 2, allowing cohort 3 escalation to sitravatinib 70 mg daily in combination with nivolumab 3 mg/kg and ipilimumab 0.7 mg/kg. Patients in cohort 3 also did not demonstrate any DLTs or other notable irAEs thus allowing further dose escalation to the final planned dose level of sitravatinib 100 mg daily combined with nivolumab 3 mg/kg and ipilimumab 0.7 mg/kg (Fig. 1b). This final cohort enrolled 9 patients to ensure safety and provide more robust efficacy signals. DLTs occurred in 3 out of 9 patients in cohort 4 (Supplementary Table S1). Of these DLTs in cohort 4, there was a grade 3 immune-related myositis / myasthenia gravis (patient ID#019), whereas the second patient (ID#026) developed grade 3 hypertension related to the anti-VEGF activity of sitravatinib that met DLT criteria, as well as grade 3 immune-related pneumonitis at day 64 which rapidly resolved with high dose steroids and therefore did not meet DLT criteria. The third patient (ID#029) developed persistent grade 2 transaminase elevation possibly related to trial therapy. There were no other notable irAEs in this cohort (Supplementary Table S1).

**Adverse events**

Study drug dose interruptions, reductions, or discontinuations by trial cohort due to treatment-related adverse events (AEs) are listed in Supplementary Table S2. Treatment-related AEs requiring discontinuation of sitravatinib throughout the treatment course were noted in 1 out of 22 patients (4.5%; grade 3 aortic injury in patient #028). Treatment-related AEs leading to dose reduction or

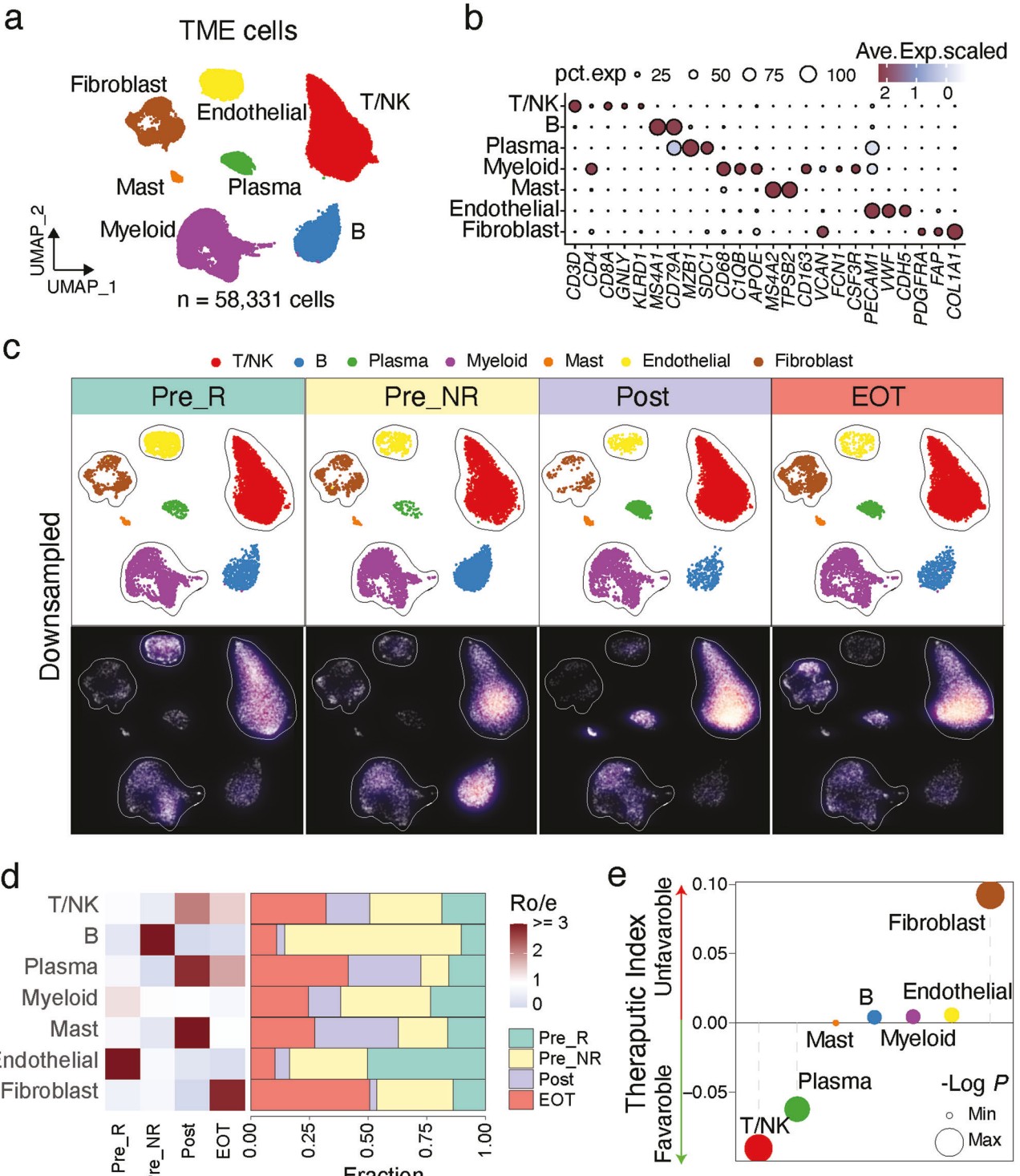

**Fig. 3 | Single-cell landscape of tumor microenvironment (TME) cells from longitudinally collected tumor samples at baseline and during trial therapy.** **a** Uniform manifold approximation and projection (UMAP) of scRNA-seq data from TME cells. Colors and numbers indicate scRNA-seq clusters. **b** Expression levels and frequencies of selected markers across TME cell clusters. Bubble size is proportional to the percentage of cells expressing a gene and color intensity is proportional to average scaled gene expression. **c** UMAP view of TME cell clusters (top) and cell density (bottom) displaying TME cell distribution across four groups. Pre_R (*n* = 3; baseline samples from patients that demonstrated an objective response to trial therapy), Pre_NR (*n* = 9; baseline samples from patients that did not demonstrate an objective response to trial therapy), Post (*n* = 4; samples collected at the

C2 timepoint), and EOT (*n* = 3; samples collected at the EOT timepoint). Downsampling was applied and 8,346 cells were included for each group. High relative cell density is shown as bright magma. **d** Distribution of TME cell clusters across groups. Left bar plot showing the relative proportion of cells from all patients for each TME cell subset. Heat map showing tissue prevalence estimated by the ratio of observed to expected cell number (Ro/e). **e** Therapeutic index of major immune cell types in ccRCC tumors treated with sitravatinib + nivolumab + ipilimumab. Dot size represents the significance evaluated by −Log10 (*p*-value). The two-sided *p*-value is obtained from the Pearson correlation analysis between cell type proportions and objective response rates without adjustment for multiple comparisons. Source data are provided in the Source Data file.

interruption of sitravatinib were noted in 17 out of 22 patients (77.3%), with 6 out of 7 belonging to cohort 1 (85.7%), 3/3 belonging to cohort 3 (100%) and 8 out of 9 belonging to cohort 4 (88.9%). Treatment-related AEs requiring discontinuation of ICT with nivolumab and ipilimumab were noted in 9/22 patients (40.9%) with 6 of these patients belonging to cohort 1 as described above and 3 of these patients belonging to cohort 4 (Grade 3 myositis in patient #019, Grade 3 hepatitis in patient #021, and Grade 3 pneumonitis in patient #026). Treatment-related AEs are listed in Supplementary Data files 1–7. Febrile neutropenia was the only sitravatinib-related grade 4 AE and occurred in one patient enrolled in cohort 4. The most common sitravatinib-related grade 3 AE was hypertension in 4/22 patients (18.2%), of which 3 patients were enrolled in cohort 1 and one patient was enrolled in cohort 4. High-dose glucocorticoids [≥40 mg/day (prednisone) or equivalent] for the management of immune-related AEs were administered in 9 of the 22 patients (40.9%) with 6 of these patients belonging to cohort 1. The median duration of steroid use for immune-related AEs was 68 days with minimum of 5 days and maximum of 132 days (Supplementary Tables S3). Additional immunosuppressive strategies such as vedolizumab, infliximab, rituximab, or plasmapheresis were needed in 6 of the 22 patients (27.3%), with 5 of those belonging to cohort 1.

## Clinical efficacy

All 22 patients enrolled were evaluable for efficacy. Overall, the combination of sitravatinib with nivolumab plus ipilimumab yielded an objective response rate (ORR) of 45.5% (10 of the 22 patients) and a disease control rate of 86.4% (19 of the 22 patients) (Fig. 2a). Clinical activity was durable (Fig. 2b) corresponding to a median progression-free survival (PFS) of 14.5 months (Fig. 2c). While the median overall survival was not reached, the 1-year survival was 80.8%, and 16/22 patients were alive at the time of data cutoff (Fig. 2d). The median duration of response (DOR) was not reached with 3/10 patients that achieved an objective response experiencing subsequent disease progression at the time of data cutoff. Clinical efficacy by dose cohort is shown in Supplementary Table S4. The recommended phase 2 dose for the triplet combination used in cohort 4 yielded ORR of 33% (3 of the 9 patients), DCR of 88.8% (8 of the 9 patients), and 1-year survival of 87.5%, while the median DOR, PFS, and overall survival were not reached, and 8/9 patients were alive at the time of data cutoff after a median follow-up time of 12.6 months.

## Sitravatinib pharmacokinetics

All 22 patients were systemically exposed to sitravatinib following oral administration. Sitravatinib reached steady state by Day 15, and that exposure increased in an approximately dose-dependent manner between the different cohorts (Supplementary Fig. S1). At Cycle 1 Day 15, the arithmetic mean pre-dose concentrations ($C_{trough}$) were 22.8 ng/mL, 19.5 ng/mL, 50.9 ng/mL, and 77.4 ng/mL for Cohorts 1, 2, 3, and 4, respectively; and at Cycle 2 Day 1 the arithmetic mean $C_{trough}$ values were 29.3 ng/mL, 20.9 ng/mL, 62.6 ng/mL, and 42.7 ng/mL for Cohorts 1, 2, 3, and 4, respectively (Supplementary Data file 8).

## Longitudinal single-cell transcriptomic landscapes

We performed single cell RNA sequencing (scRNA-seq) in 19 tissue samples collected from 12 patients enrolled in the trial and treated in **cohorts 2, 3 and 4** (Supplementary Table S5). Baseline samples were collected from all 12 patients (samples from 9 patients coded as "Pre_NR" who did not demonstrate an objective response to trial therapy, and from 3 patients coded as "Pre_R" who showed an objective response to trial therapy), while 4 patients also had samples collected prior to the second infusion of nivolumab plus ipilimumab (C2 timepoint; also termed as "Post") and 3 patients also had samples at the end of treatment (EOT timepoint) due to disease progression. Whenever possible, longitudinally collected samples were biopsied from the same organ site. All three timepoints were collected from the same

tumor site in two patients, ID #016 and 018, while patient ID #020 had baseline and EOT samples collected (Supplementary Table S5). The scRNA-seq captured transcriptomes from approximately 70,526 cells following stringent quality control (see Methods), comprised of 58,331 cells from the TME (Fig. 3a) and 12,195 epithelial cells (Supplementary Fig. S2a). Unsupervised clustering was performed on TME cells and epithelial cells separately to identify distinct cell types and transcriptional states. As reported in many other scRNA-seq studies[10–12], immune cells and stromal cells were clustered by cell type (Supplementary Fig. S2b, c), while the epithelial cell clusters exhibited distinct patient and tissue specificity (Supplementary Fig. S2a–c).

Epithelial cells were grouped into 21 clusters by unsupervised clustering (Supplementary Fig. S2d). We then applied the inferCNV algorithm to estimate copy number variations (CNVs) in epithelial cells using stromal cells of this study as the control (Supplementary Fig. S2d, e). Based on inferred CNVs and clustering analysis (Supplementary Fig. S2d–f), epithelial cells were subsequently classified into malignant cell clusters and normal epithelial cell clusters (Supplementary Fig. S2f).

We first characterized the TME cells. Unsupervised clustering analysis identified 8 major types of TME cells (Fig. 3a), including B cells (*MS4A1* and *CD79A*), T cells (*CD3D, CD4*, and *CD8A*), NK cells (*GNLY* and *KLRD1*), plasma cells (*MZB1* and *SDC1*), myeloid cells (*CD68* and *C1QB, APOE*), mast cells (*MS4A2* and *TPSB2*), endothelial cells (*PEACAM1* and *VWF*), and fibroblasts (*OL1A1* and *FAP*), each exhibits unique expressing profiles (Fig. 3b). Based on their sample origin, we divided the TME cells into four groups: Pre_R (baseline samples from patients that demonstrated an objective response to trial therapy), Pre_NR (baseline samples from patients that did not demonstrate an objective response to trial therapy), Post (samples collected at the C2 timepoint), and EOT (samples collected at the EOT timepoint), and observed substantial differences in cell compositions across these groups (Fig. 3c). Notably, when comparing the Post with Pre_R TME cells, there was a trend towards increased fractions of T cells and slightly decreased fractions of myeloid cells (Supplementary Fig. S2g). Consistent with the anti-angiogenic effect of sitravatinib[13], a significant decrease in endothelial cells was noted throughout the treatment course, including at EOT, based on the ratio of observed to expected cell numbers (Ro/e) (Fig. 3c, d). The same patterns were observed when comparing only the subset of paired baseline and EOT samples obtained longitudinally from the same patients (Supplementary Fig. S7a) or only patients treated in cohort 4 (Supplementary Fig. S8a).

We next computed the therapeutic index as previously described to systematically explore the relationship between distinct immune cell types and clinical responses[14]. This index quantifies the associations between changes in cellular proportions and tumor size alterations. A negative therapeutic index indicates that a higher baseline level of cellular proportion of the corresponding immune cell subtype is linked to a greater decrease in tumor size after treatment, thus predicting a favorable response. Conversely, a positive therapeutic index suggests that the respective immune cell subtype predicts an unfavorable response. In this study, T cell abundance at baseline correlated with a better objective treatment response, whereas fibroblast abundance at baseline correlated with a worse objective treatment response (Fig. 3e), consistent with recent findings regarding the negative predictive roles of fibroblasts in ICT response[15]. A sensitivity analysis using only the subset of patients treated in cohort 4 showed similar findings (Supplementary Fig. S8b).

## EMT-like signature associated with treatment resistance

As observed in many other studies[10–12], tumor cells exhibit a high degree of inter- and intra-patient transcriptional heterogeneity (Fig. S2a, b). To better understand how tumor cell states at baseline and their evolution during therapy can influence treatment resistance, we employed Harmony[16] for batch correction and the removal of inter-

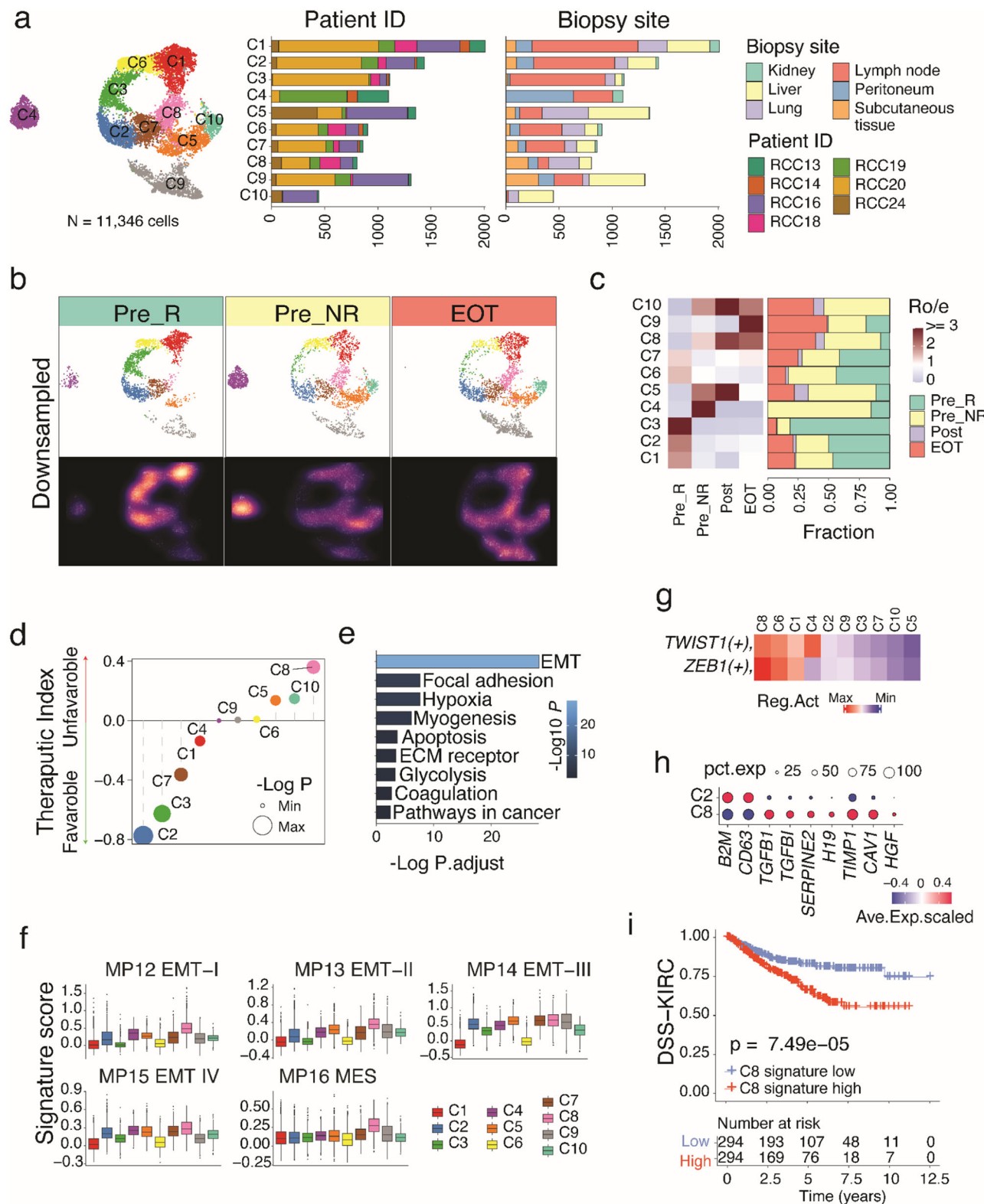

patient transcriptional heterogeneity, aiming to identify cancer cell states and malignant programs that are shared across patients. With Harmony, malignant epithelial cells were clustered into 10 clusters, exhibiting no organ- or patient-specificity, which is indicative of Harmony's good performance in the removal of inter-patient transcriptional heterogeneity (Fig. 4a and Supplementary Fig. S3a). Among these cell clusters, C1, C2, C3 and C7 were significantly enriched in pretreatment samples that exhibited favorable objective responses to

trial therapy. The higher the proportions of these cell clusters, the more pronounced the response to treatment. The effect was particularly notable in C2 tumor cells, which demonstrated the lowest positive therapeutic index (Fig. 4b–d). Conversely, C5, C8 and C10 groups showed the opposite trend, i.e., their cellular proportions increased after drug treatment and higher baseline levels were associated with poor objective response to trial therapy (Fig. 4b–d). Specifically, C8 tumor cells were enriched in Pre_NR and EOT samples (Fig. 4c),

**Fig. 4 | EMT high tumor cell clusters associated with treatment resistance.**
**a** Uniform manifold approximation and projection (UMAP) of scRNA-seq data from malignant cells. Colors and numbers indicate scRNA-seq clusters. Left bar plot showing the relative proportion of cells from all patients and biopsy sites for each malignant cell subset. **b** UMAP view of malignant cell clusters (top) and cell density (bottom) displaying malignant cell distribution across three groups. Pre_R ($n = 3$; baseline samples from patients that demonstrated an objective response to trial therapy), Pre_NR ($n = 9$; baseline samples from patients that did not demonstrate an objective response to trial therapy), and EOT ($n = 3$; samples collected at the EOT timepoint). Downsampling was applied and 2597 cells were included for each group. High relative cell density is shown as bright magma. **c** Distribution of malignant cell clusters across groups. Left bar plot showing the relative proportion of cells from all patients for each malignant cell subset. Heat map showing tissue prevalence estimated by the ratio of observed to expected cell number (Ro/e).
**d** The therapeutic index of major immune cell types in ccRCC tumors treated with sitravatinib + nivolumab + ipilimumab. Dot size represents the significance evaluated by −Log10 (*p*-value). The two-sided *p*-value is obtained from the Pearson correlation analysis between cell type proportions and objective response rates

without adjustment for multiple comparisons. **e** Pathway enrichment analysis for signature genes in C8 cell clusters. Barplot shows selected top ranked pathways. The *p*-value was calculated using a two-sided Fisher's exact test without adjustment for multiple comparisons. **f** Expression levels of selected EMT related meta-program signatures across malignant cell clusters. $N = 2011$ cells for C1, 1437 cells for C2, 1113 cells for C3, 1101 cells for C4, 1355 cells for C5, 904 cells for C6, 860 cells for C7, 806 cells for C8, 1313 cells for C9, and 446 cells for C10. Box-and-whisker plots show all values with range (whiskers), interquartile range (box) and median (center line). **g** Selected EMT transcription factor activities between different malignant cell clusters. **h** Bubble plot showing key marker gene expression between the C2 and C8 cell clusters. **i** Disease-specific survival (DSS) differences based on C8 signature expression level in patients with clear cell renal cell carcinoma (KIRC) in the TCGA dataset. The *p*-value was calculated using the log-rank test to compare survival distributions between the defined groups. A two-sided test without adjustment for multiple comparisons was applied to assess whether there were significant differences in DSS between the groups. Source data are provided in the Source Data file.

---

showing the highest positive therapeutic index, indicating that a higher baseline proportion of C8 tumor cells predicts the most unfavorable response to treatment with sitravatinib plus nivolumab plus ipilimumab (Fig. 4d). The enrichment for C8 cells at EOT was also observed when comparing only the subset of paired baseline and EOT samples obtained longitudinally from the same patients (Supplementary Fig. S7b–d). We further quantified and compared the expression levels of tumor hallmark and metabolic pathways of these two tumor cell clusters. Compared to other cell clusters, C8 exhibited EMT, hypoxia, myogenesis, and glycolysis signal activity (Fig. 4e). Consistently, further analysis of 41 previously defined cancer cell meta-programs (MPs)[17] suggested that all EMT-related MPs activities were elevated in C8 tumor cells. Particularly, C8 tumor cells exhibited the highest expression of MP12 EMT-I, MP13 EMT-II, and MP16 mesenchymal (MES) programs (Fig. 4f and Supplementary Fig. S3b). In line with this, transcription factor activity analysis revealed significantly higher activities of EMT-related transcription factors, such as ZEB1 and TWIST1[18], in C8 tumor cells (Fig. 4h and Supplementary Fig. S3c). Conversely, C2 tumor cells displayed a higher expression level of B2M, which is responsible for the stabilization of MHC class I molecules at the cell surface and is associated with immunotherapy sensitivity in clear cell RCC[19] (Fig. 4g). A sensitivity analysis focusing only on the subset of patients enrolled in cohort 4 produced similar findings as the main analysis (Supplementary Fig. S8c–g). Furthermore, tumor tissues with sarcomatoid and/or rhabdoid dedifferentiation, known to be associated with aggressive disease course following progression on ICT[20,21], were significantly enriched for C8 tumor cells (Supplementary Fig. S9a–d).

To further assess the clinical significance of C8 tumor cells, we extracted differentially expressed genes from C8 tumor cells and constructed a C8 tumor cell gene signature (Supplementary Data file 9). Based on computed gene signature scores from this signature, we categorized tumor cells into high- and low-signature groups based on group median levels (Supplementary Fig. S3d). Notably, the signature activity derived from C8 was associated with poor patient treatment responses and showed a negative correlation with progression-free survival during trial therapy (Supplementary Fig. S3e, f). Moreover, we externally validated these findings in the TCGA kidney renal clear cell carcinoma (KIRC) cohort and found a significant association ($p < 0.001$) between the C8 tumor cell signature activity and disease-specific survival in patients with ccRCC (Fig. 4i). This strong signal persisted in multivariable analysis adjusting for tumor stage and history of neoadjuvant therapy (Supplementary Fig. S3g and Supplementary Table S6). Similar findings were noted in both unadjusted and adjusted analysis of the association between C8 tumor cell signature

activity and overall survival in the TCGA KIRC cohort (Supplementary Fig. S3h, i and Supplementary Table S7).

## Longitudinal T cell changes during trial therapy

We identified 10 different CD4 + T cell states (Fig. 5a), including naïve CD4 T cells (Tn, c0 and c3), follicular helper T cells (Tfh, c6), central memory T cells (Tcm, c1), regulatory T cells (Treg, c2 and c5), cytotoxic T cells (CTL, c4, c7 and c8), and proliferative T cells (Tprolif, c9). Consistently with our previous pan-cancer studies[11], the Tn clusters expressed high levels of the naïve gene signature (Fig. 5d and Supplementary Fig. S4a, d). The CTL clusters markedly expressed cytolytic activity-related genes and chemokine/chemokine receptors (Fig. 5d and Supplementary Fig. S4a, c, d). The Treg cluster exhibited a classical Treg gene signature (*TIGIT*, *FOXP3*, *CTLA4* and *TNFRSF4*) (Supplementary Fig. S4a, b). The Tfh cluster was characterized by high expression of *ICOS*, *TCF7*, *TOX*, *CXCL13* and *PDCD1*. The Tprolif cluster was marked by cell cycle-related markers (Fig. 5d). CD4 + T cell states and compositions varied significantly between different timepoints (Fig. 5c, d and Supplementary Fig. S4c). CTL subsets were highly abundant in Pre_R but decreased in Pre_NR and EOT tissues (Fig. 5b, c). These CTL subsets were also enriched in ccRCC tissues with sarcomatoid and/or rhabdoid dedifferentiation (Supplementary Fig. S9e, f), consistent with their established sensitivity to ICT[20]. Treg subsets were lower in baseline tissues but abundant in EOT tissues. Similar patterns were observed when comparing only the subset of paired baseline and EOT samples obtained longitudinally from the same patients (Supplementary Fig. S7e, f). However, we did not observe enrichment of Treg subsets in EOT tissues when comparing only patients enrolled in cohort 4 (Supplementary Fig. S8h, i). In summary, the abundance of cytotoxic CD4 + T cells correlates with a better prognosis in patients. However, during later stages of treatment, some patients exhibit increased levels of Treg cells (Fig. 5e), which are associated with poor ICT outcomes[11].

Unsupervised clustering analysis identified 8 CD8 + T cell transcriptional states (Fig. 5f): Tn (c5), effector (Teff, c0, c3 and c4), exhausted (Tex, c1), proliferative (Tprolif, c7) and two unconventional CD8 + T cell states. The Tn cluster displayed a naïve-like phenotype with high expression of a naïve gene signature (Fig. 5h and Supplementary Fig. S5a, d). The three Teff clusters highly expressed effector molecules (e.g., *FGFBP2*, *CX3CR1*, *GZMK* and GZMA), cytolytic activity-related genes and consistently, high cytotoxicity gene signature and T cell receptor (TCR) signaling (Fig. 5h and Supplementary Fig. S5a, b, d). The Tex cluster was characterized by the highest expression of exhaustion-related gene signature (e.g., *LAG3*, *PDCD1*, *TIGIT*, *HAVCR2*, *TOX* and *CTLA-4)*, with lower expression of *TCF7* and cytolytic activity-

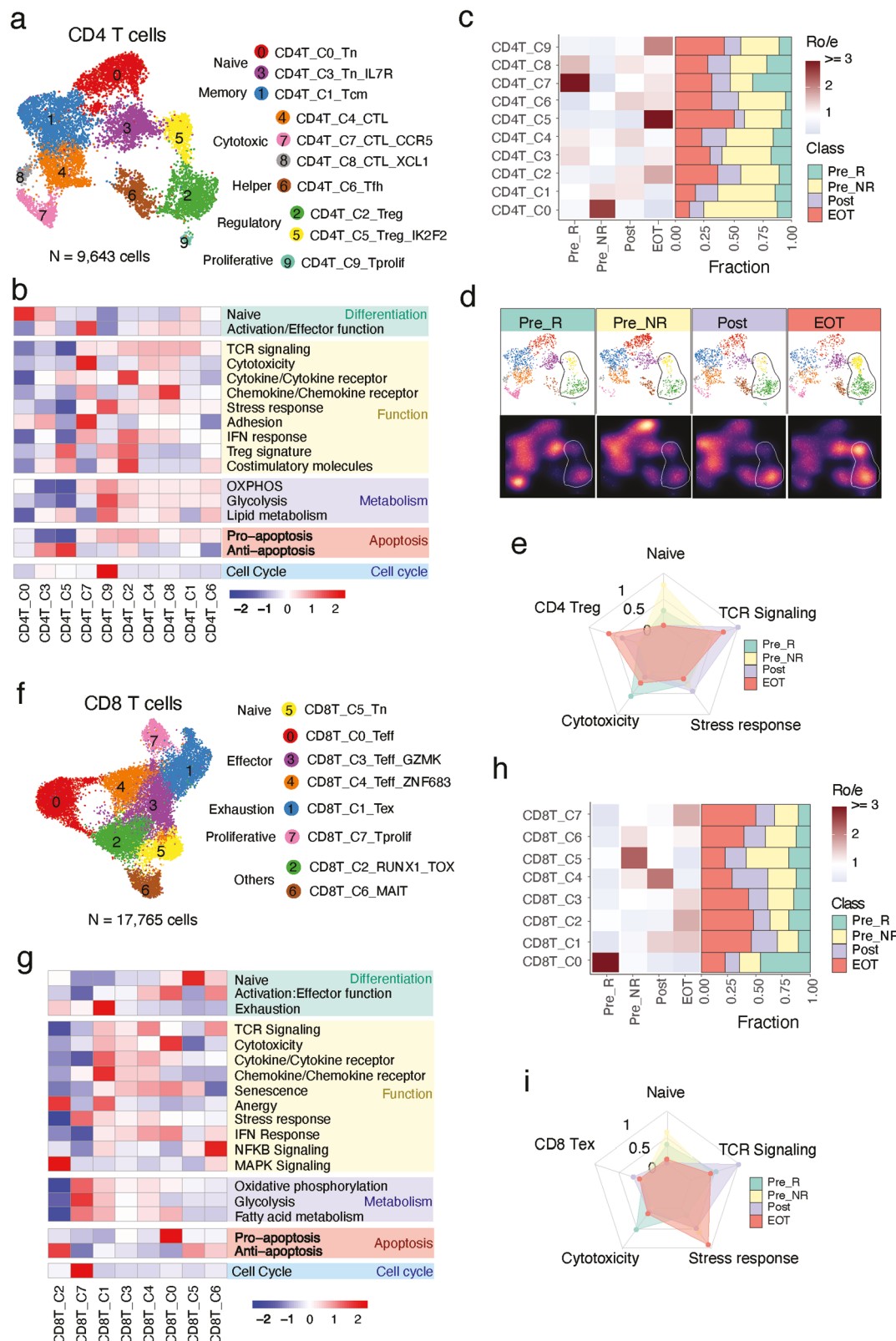

related genes (e.g., *GZMB, GZMH, PRF1*) (Fig. 5h and Supplementary Fig. S5a). The Tprolif cluster was marked by cell cycle-related markers. The states and compositions of CD8 + T cells showed notable variations across treatment timepoints (Fig. 5g and Supplementary Fig. S5c). The Teff subset C0 cells were prevalent in Pre_R tissues but were notably reduced in Pre_NR and EOT tissues. Conversely, the Tex subset was relatively more abundant in EOT tissues compared to the

baseline tissues (Fig. 5g and Supplementary Fig. S5c). This pattern was similarly observed when analyzing the subset of paired baseline and EOT samples collected longitudinally from the same patients (Supplementary Fig. S7g, h). In addition, we observed a significant increase in the stress response state of CD8 + T cells in Pre_NR and EOT tissues (Fig. 5i). This stress state is associated with immunotherapy resistance[11] and may contribute to poor treatment responses in these patients.

**Fig. 5 | Longitudinal T cell changes during trial therapy. a** Uniform manifold approximation and projection (UMAP) of scRNA-seq data from CD4 + T cells. Colors and numbers indicate scRNA-seq clusters. **b** Distribution of CD4 + T cell clusters across groups. Left bar plot showing the relative proportion of cells from all patients for each CD4 + T cell subset. Heat map showing tissue prevalence estimated by the ratio of observed to expected cell number (Ro/e). Pre_R (*n* = 3; baseline samples from patients that demonstrated an objective response to trial therapy), Pre_NR (*n* = 9; baseline samples from patients that did not demonstrate an objective response to trial therapy), Post (*n* = 4; samples collected at the C2 timepoint), and EOT (*n* = 3; samples collected at the EOT timepoint). **c** UMAP view of CD4 + T cell clusters (top) and cell density (bottom) displaying CD4 + T cell distribution across four groups. Downsampling was applied and 1,345 cells were included for each group. High relative cell density is shown as bright magma. **d** Heat map illustrating expression of 17 curated gene signatures across CD4 + T cell clusters. Heat map was generated based on the scaled gene signature scores. **e** Radar plot showing enrichment of selected five CD4 + T cell states. **f** UMAP of scRNA-seq data from CD8 + T cells. Colors and numbers indicate scRNA-seq clusters. **g** Distribution of CD8 + T cell clusters across groups. Left bar plot showing the relative proportion of cells from all patients for each CD8 + T cell subset. Heat map showing tissue prevalence estimated by Ro/e. **h** Heat map illustrating expression of 20curated gene signatures across CD8 + T cell clusters. Heat map was generated based on the scaled gene signature scores. **i** Radar plot showing enrichment of selected five CD8 + T cell states. Source data are provided in the Source Data file.

However, these patterns were not observed when evaluating only the subset of patients enrolled in cohort 4 (Supplementary Fig. S8j, k). Conversely, ccRCC tissues with sarcomatoid and/or rhabdoid dedifferentiation were enriched for cytotoxic CD8 + T cells, whereas tissues without sarcomatoid or rhabdoid components harbored significantly higher CD8 + T cells in the stress response state (Supplementary Fig. S9g, h).

### Longitudinal myeloid cell changes during trial therapy

We identified 12 myeloid cell states, including 5 tumor-associated macrophage (TAM) clusters, 1 mast cell cluster, and 4 dendritic cell (DC) clusters (Fig. 6a and Supplementary Fig. S6a). The TAM_C0 cluster expressed lipid metabolism genes (*APOC1*, and *APOE*). TAM_C2 showed high expression of non-classical myeloid marker genes, such as *SELENOP*, *FOLR2*, and *C1QA*, associated with the resident macrophage type[22]. TAM_C3 exhibited high expression of *LRMDA*, *TCF12*, and other MT-related genes. TAM_C4 highly expressed canonical marker gene *CD163* and one sitravatinib targeting gene *MERTK*. TAM_C8 was defined by cell cycle-related markers. In addition, the DC subsets included pDC, the classical CLEC9A + cDC1 (c10), CD1C + cDC2 (c6), and a LAMP3+ mature cDC subset (c11) (Fig. 6a and Supplementary Fig. S6a).

TAMs and DCs were predominantly present in Pre_NR and EOT tissues, while monocytes mainly originated from Pre_R tissues (Fig. 6b, c and Supplementary Fig. S6b). Overall, TAMs showed high expression of the phagocytosis gene signature (Fig. 6d and Supplementary Fig. S6c, d). TAM clusters also highly expressed M2-like, angiogenesis-related gene signatures, and inhibitory immune checkpoint genes (Supplementary Fig. S6c), suggesting an immunosuppressive phenotype. cDC1, exhibiting the lowest expression of the M2-like gene signature and the highest expression of the antigen-presenting cell (APC) gene signature, was less abundant in baseline samples but dominant in EOT tissues (Fig. 6b, c and Supplementary Fig. S6b). LAMP3+ DCs, displaying the highest expression of *CD274* (*PD-L1*), *IDO1*, and *TIGIT* signaling genes, were enriched in EOT tissues (Fig. 6b, c and Supplementary Fig. S6b). We further identified markedly distinct patterns of cell communications (Fig. 6e). The malignant cell clusters (C8, C5, and C10) associated with poor treatment responses (Fig. 4d) interacted with TAMs through the CSF1-CSF1R receptor, whereas such interactions were not observed in malignant cell clusters positively correlated with response (C2, C3 and C7) (Fig. 6e). EOT and Pre_NR tissues harbored significantly higher levels of M2-like myeloid cells overall (Fig. 6f), expressing significantly higher levels of *CSF1R* and genes related to IL6-JAK-STAT3 signaling (Fig. 6g, h and Supplementary Fig. S6e, f). A sensitivity analysis focusing only on the subset of patients enrolled in cohort 4 showed similar findings (Supplementary Fig. S8l, m). However, this pattern was not evident when examining the subset of paired baseline and EOT samples collected longitudinally from the same patients (Supplementary Fig. S7i–l). Tissues from ccRCC with sarcomatoid and/or rhabdoid dedifferentiation expressed significantly higher levels of *CSF1R* and genes related to IL6-JAK-STAT3 signaling but were not enriched for M2-like myeloid cells (Supplementary Fig. S9j–l). Overall, myeloid cells exhibited considerable abundance across diverse lineages, characterized by a spectrum of transcriptional states and altered cellular compositions. As treatment resistance was acquired, these myeloid cells transitioned from immune-stimulating to immunosuppressive phenotypes (Fig. 6b, c and Supplementary Fig. S6b), indicative of a dynamic interplay within the TME that may influence therapeutic outcomes.

### Discussion

In contrast to the TKI-based triplet immunotherapy combination of cabozantinib with nivolumab plus ipilimumab used in ccRCC[23,24], the present study was unable to safely combine sitravatinib with the nivolumab 3 mg/kg plus ipilimumab 1 mg/kg ICT backbone. This may be due to the distinct immunomodulatory properties of sitravatinib[6,25,26]. We found no evidence that sitravatinib pharmacokinetics were influenced by ipilimumab, and sitravatinib steady-state exposures in cohort 4 of our trial were similar to those observed in a previous study investigating sitravatinib and nivolumab doublet combination in patients with ccRCC at the recommended phase 2 dose of 100 mg daily[26].

Many phase 1 trials use a short DLT window for pragmatic reasons because accounting for late-onset toxicities can substantially increase the time it takes for the trial to be completed[27,28]. To address this challenge, we used a time-to-event Bayesian optimal interval (TITE-BOIN) design that is well-suited for use in dose escalation studies involving treatments associated with late-onset toxicity[29]. Indeed, most of the serious irAEs requiring steroids in our study occurred within the 9-week DLT window. Per protocol, grade 3 irAEs that resolved to Grade ≤1 or baseline with immunosuppressive therapy within 3 weeks were not counted as DLTs. Most irAEs treated with immunosuppressives in our trial resolved within this 3-week timeframe and therefore were not considered DLTs. However, the irAE management was performed at a high-volume tertiary referral center and would be difficult to replicate in the community[30]. Therefore, the overall frequency and challenging quality of irAEs observed in 6 out of 7 patients treated with the triplet therapy in cohort 1 made the combination of sitravatinib with the nivolumab 3 mg/kg plus ipilimumab 1 mg/kg ICT backbone a non-viable option pragmatically. However, the strong early evidence of efficacy compelled additional investigation of strategies to more safely combine sitravatinib with nivolumab plus ipilimumab without compromising clinical activity. Longer follow up will be needed to assess potential long-term risks to patients at all treated cohorts given the high immunogenicity of sitravatinib.

The irAEs in cohort 1 would often occur after the first 1, 2 doses of nivolumab plus ipilimumab. Therefore, reducing the total lifetime number of ipilimumab doses to less than four or lengthening the time interval to longer than every 3 weeks would be less likely to meaningfully improve the toxicity profile of the triplet. On the other hand, previous studies have suggested that lowering the dose of ipilimumab to less than 1 mg/kg can substantially decrease toxicity while maintaining efficacy[8,9]. Indeed, dose reduction of ipilimumab to 0.7 mg/kg in combination with nivolumab 3 mg/kg substantially improved the

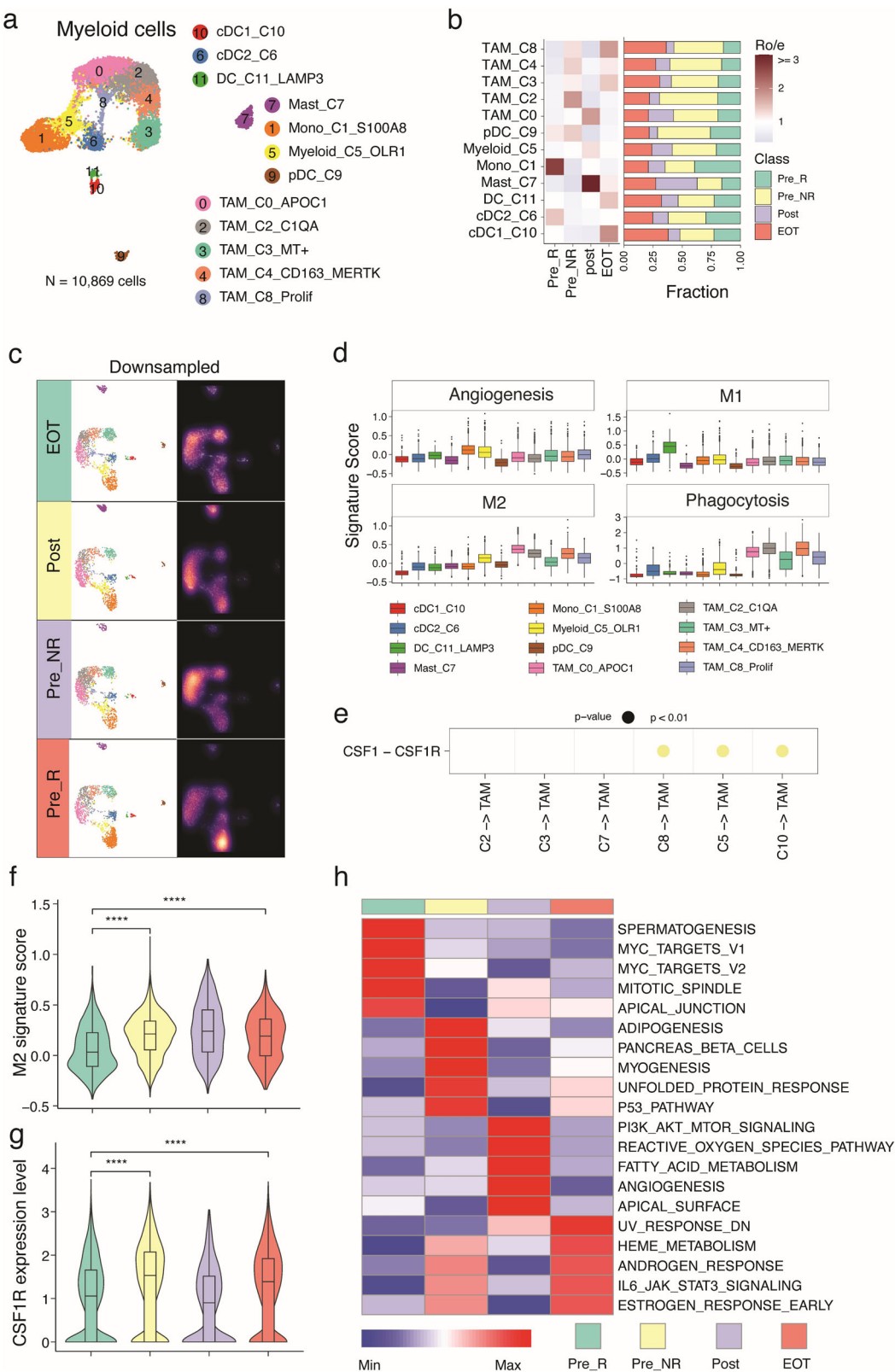

toxicity profile of the triplet and allowed safe dose escalation of sitravatinib to the final planned dose level of 100 mg daily in cohort 4. This is typically the maximum sitravatinib dose that is well tolerated as either monotherapy or in combination with nivolumab[26]. Sitravatinib was well tolerated with febrile neutropenia being the only sitravatinib-related grade 4 AE occurring in one patient enrolled in cohort 4. Consistent with prior experience of sitravatinib in ccRCC[26], the most

common sitravatinib-related grade 3 AE was hypertension that was manageable with medical therapy. This tolerability enabled escalation to the maximum dose level of sitravatinib which further distinguishes this triplet regimen from the combination of cabozantinib with nivolumab 3 mg/kg and ipilimumab 1 mg/kg which uses cabozantinib doses of 40 mg daily, whereas the monotherapy starting dose is 60 mg daily[23,24,31]. While the signal of clinical benefit and durable responses is

**Fig. 6 | Longitudinal myeloid cell changes during trial therapy. a** Uniform manifold approximation and projection (UMAP) of scRNA-seq data from myeloid cells. Colors and numbers indicate scRNA-seq clusters. **b** Distribution of myeloid cell clusters across groups. Left bar plot showing the relative proportion of cells from all patients for each myeloid cell subset. Heat map showing tissue prevalence estimated by the ratio of observed to expected cell number (Ro/e). Pre_R (*n* = 3; baseline samples from patients that demonstrated an objective response to trial therapy), Pre_NR (*n* = 9; baseline samples from patients that did not demonstrate an objective response to trial therapy), Post (*n* = 4; samples collected at the C2 timepoint), and EOT (*n* = 3; samples collected at the EOT timepoint). **c** UMAP view of myeloid T cell clusters (left) and cell density (right) displaying myeloid cell distribution across four groups. Downsampling was applied and 1601 cells were included for each group. High relative cell density is shown as bright magma. **d** Expression levels of pathway signatures across myeloid cell clusters. *N* = 1950 cells for TAM_C0_APOC1, 1914 cells for Mono_C1_S100A8, 1725 cells for TAM_C2_C1QA, 1253 cells for TAM_C3_MT + , 1047 cells for TAM_C4_CD163_MERTK, 898 cells for Myeloid_C5_OLR1, 593 cells for cDC2_C6, 529 cells for Mast_C7, 497 cells for TAM_C8_Prolif, 208 cells for pDC_C9, 171 cells for cDC1_C10, and 84 cells for DC_C11_LAMP3. Box-and-whisker plots show all values with range (whiskers), interquartile range (box) and median (center line). **e** Cell communication analysis

reveals strong CSF1-CSF1R cell communication between malignant cell populations leading to treatment resistance and TAMs. The *p*-values represent the likelihood of observing the identified or more extreme interactions if the null hypothesis of no interactions was correct, with Benjamini-Hochberg adjusted *p*-values for multiple testing correction. **f** Expression levels of M2 macrophage related signature in macrophage and monocytes cell clusters across groups. *N* = 2517 cells for Pre_R, 4074 cells for Pre_NR, 1601 cells for Post, and 2677 cells for the EOT timepoint. Box-and-whisker plots show all values with range (whiskers), interquartile range (box) and median (center line). ****p*-value < 0.0001. The *p*-value was calculated by using the two-sided Wilcoxon rank-sum test without adjustment for multiple comparisons. **g** Expression levels of *CSF1R* gene in macrophage and monocytes cell clusters across groups. *N* = 2517 cells for Pre_R, 4074 cells for Pre_NR, 1601 cells for Post, and 2677 cells for the EOT timepoint. Box-and-whisker plots show all values with range (whiskers), interquartile range (box) and median (center line). ****p*-value < 0.0001. The *p*-value was calculated by using the two-sided Wilcoxon rank-sum test without adjustment for multiple comparisons. **h** Heat map illustrating expression of top 5 selected hallmark signatures across groups in macrophage and monocytes cell clusters. Heat map was generated based on the scaled gene signature scores. Source data are provided in the Source Data file.

encouraging, further investigation would be needed in a larger number of patients and with longer follow-up to reliably establish the efficacy of the recommended phase 2 dose used in cohort 4.

To investigate the effects of the triplet therapy in the TME and malignant cells at the single-cell level, we performed scRNA-seq in longitudinally collected samples. This allowed the identification of resistant tumor cells characterized by upregulation of EMT and hypoxia pathways. Resistant cells to the triplet regimen are refractory to the two systemic therapy mainstays of ccRCC management: ICT and anti-VEGF tyrosine kinase inhibitors (TKIs)[1,32]. Indeed, we found that the gene signatures derived from these cells were also associated with shorter survival in patients with ccRCC within the TCGA dataset. These transcriptomic hallmarks can inform future efforts to develop therapeutic strategies tailored towards eradicating or preventing the emergence of these highly resistant ccRCC cells. As expected given the anti-VEGF activity of sitravatinib, we found that endothelial cells decrease following initiation of the triplet therapy. Notably, endothelial cells were not upregulated upon emergence of treatment resistance, consistent with the observation that resistant tumor cells engage pathways driven by hypoxia[33,34].

Sitravatinib alone and in combination with ICT has been previously shown to increase CD3+ positive T cells in the TME[6]. We indeed observed an increase in CD3 + T cells in the TME following triplet therapy initiation. Notably, prior combinations of ICT with TKIs were also based on the preclinical rationale that the targeted receptor tyrosine kinase pathways can induce a significant positive immunomodulatory effect[25]. However, clinical results with triplet combinations have been disappointing although other TKIs have not shown the increased irAEs noted with sitravatinib suggestive of a potent immunomodulatory effect[23,24,35]. We also noted that emergence of treatment resistance was characterized by the transition from cytotoxic to exhausted T cells. This was accompanied by enrichment for M2-like myeloid cells with biological signatures corresponding to a resolution-phase or resident type macrophage phenotype[22,36]. Further investigation will be needed to determine whether reversal or prevention of the observed longitudinal TME changes associated with triplet therapy resistance can improve outcomes of patients with ccRCC and other malignancies.

We found that high T/NK cell infiltration was associated with favorable response to the triplet combination. Although high NK cell infiltration is known to be associated with an improved prognosis in ccRCC[37], increased CD8 + T cell infiltrates has been linked to worse survival in patients with ccRCC[38]. Recent work suggests that specific

subsets of dysfunctional tumor-infiltrating CD8 + T cells may be driving the worse prognosis[39]. We indeed noted a significant association with the stress response state of CD8 + T cells in tissues of non-responders to sitravatinib in combination with ICT. Additional investigation will be needed to elucidate how specific T cell subsets contribute to the differential responses to treatment in ccRCC.

Limitations of this single-centre phase 1 trial include the possibility of self-selection bias whereby patients willing to participate may have more motivation, better access to healthcare, or be healthier than the general eligible population, potentially skewing results towards better outcomes. Other biases included the possibility of referral bias whereby referrals from oncologists might prioritize patients more likely to benefit from the trial, potentially inflating efficacy metrics. There may also be a single-center bias whereby results may not generalize to broader, more diverse populations, as participants might share regional or demographic characteristics specific to the academic centre.

In conclusion, the present study established a safe dose for combining sitravatinib with nivolumab plus ipilimumab. This may have come at the cost of reduced ORR in the recommended phase 2 dose used in cohort 4 compared with what was initially observed in cohort 1 using the standard nivolumab 3 mg/kg and ipilimumab 1 mg/kg backbone. However, the small subset of patients in cohort 4 maintained high disease control and durable PFS, which may warrant further investigation in a larger study. Notably, the longitudinal correlative analyses at the single-cell level allowed the identification of a tumor cell-specific gene signature that can predict treatment resistance and poor clinical outcomes to triplet therapy in patients with advanced ccRCC. Prospective validation of this signature may facilitate selection of patients most likely to benefit from this therapy.

## Methods
### Trial oversight
The study was approved by the institutional review board/independent ethics committee of The University of Texas MD Anderson Cancer Center (MDACC; IRB protocol 2020-0264), registered under clinicaltrials.gov identifier NCT04518046, approved by the MDACC institutional review board (IRB)/independent ethics committee and conducted in accordance with Good Clinical Practice guidelines, defined by the International Conference on Harmonisation. All patients provided written informed consent to participate based on the principles of the Declaration of Helsinki. The full trial protocol is available in the supplementary note.

## Patients

Eligible patients were ≥ 18 years old, had pathologic confirmation of advanced or metastatic, poor- or intermediate-risk clear cell renal cell carcinoma (ccRCC) with a clear cell component. Eligible patients had no prior treatment with systemic therapy with the following exception: One prior adjuvant or neoadjuvant therapy for completely resectable ccRCC was allowed if such therapy did not include an agent that targets VEGF or VEGF receptors or any other antibody or drug targeting T-cell co-stimulation or checkpoint pathways (including, but not limited to: an anti-PD-1, anti-PD-L1, anti-PDL2, anti-CD137, or anti-CTLA-4 antibody), and if recurrence occurred at least 6 months after the last dose of adjuvant or neoadjuvant therapy. Additional key inclusion criteria were measurable disease according to Response Evaluation Criteria in Solid Tumors (RECIST version 1.1) and an Eastern Cooperative Oncology Group performance status score of 0 or 1. Patients were excluded if they had brain metastases, conditions requiring corticosteroids ( > 10 mg daily of prednisone or equivalent) or other immunosuppressive medication, autoimmune disease, or history of deep vein thrombosis or pulmonary embolism within the past 6 months (unless stable and treated with appropriate anticoagulation with the exception of warfarin). Patients were permitted to discontinue from study treatment or from the study at any time at their own request, or by the discretion of the Investigator or Sponsor for safety, behavioural reasons, or for significant protocol violations. Further discontinuation reasons included objective disease progression, global deterioration of health, adverse events, loss to follow-up, refusal for further treatment, study termination by Sponsor, and death. Refer to the full trial protocol in the supplementary note for more details.

## Trial design

This was an open-label phase 1 dose-escalation study (516-008 trial) conducted at MDACC. The clinical trial was designed to evaluate the safety, clinical activity, pharmacokinetics and tissue correlatives of sitravatinib in combination with nivolumab and ipilimumab in frontline ccRCC. The triplet regimen composed of sitravatinib QD plus intravenous nivolumab Q3W and intravenous ipilimumab Q3W for four cycles before continuing maintenance nivolumab (240 mg Q2W or 480 mg Q4W) with sitravatinib daily until progressive disease (PD) or unacceptable toxicity (Fig. 1A). Whenever possible, optional fresh tumor tissue biopsies were collected at screening, cycle 2 day 1, and after PD but before other anticancer therapy.

The first cohort of patients enrolled at a sitravatinib dose of 35 mg daily with 3 mg/kg of nivolumab and 1 mg/kg of ipilimumab (NIVO3/IPI1) at the aforementioned regimen; NIVO3/IPI1 is approved for treatment of this patient population and sitravatinib 35 mg QD is well below the RP2D of sitravatinib (100 mg QD) in combination with nivolumab. Dose escalation/de-escalation decisions followed the time to event Bayesian optimal interval (TITE-BOIN) design and decision table as described in the full protocol in the supplementary note.

The trial was designed to allow for future dose expansion cohorts via protocol amendment to include other solid malignancies in which favorable activity has been previously demonstrated with nivolumab plus ipilimumab combination treatment, such as metastatic colorectal carcinoma, melanoma, hepatocellular carcinoma, non-small cell lung cancer, and urothelial carcinoma, and this is the reason behind the discrepancy between planned (up to 92) and actually enrolled patients ($n = 22$).

## Dose-limiting toxicities

DLTs were graded according to the National Cancer Institute Common Terminology Criteria for AEs (NCI-CTCAE) version 5.0. The DLT evaluation period was defined as the first 3 weeks of treatment for non-immune related AEs (sitravatinib-related AEs regardless of relationship to nivolumab and/or ipilimumab), and the first 9 weeks of treatment for immune-related AEs (nivolumab- and/or ipilimumab-related AEs, regardless of relationship to sitravatinib). The purpose of this longer window was to capture irAEs that fall within the known, late-onset toxicities for nivolumab and/or ipilimumab.

The following non-immune related AEs were designated as DLTs:
- Grade 5 AE;
- Grade ≥4 hematologic AE lasting ≥4 days;
- Grade ≥3 thrombocytopenia associated with clinically significant bleeding;
- Grade ≥3 febrile neutropenia;
- Grade ≥4 non-hematologic AE;
- Grade 3 hypertension that cannot be controlled with medical therapy, including:

  - Severe hypertension with systolic blood pressure ≥180 mmHg or diastolic blood pressure ≥120 mmHg, on more than one occasion;
  - Sustained uncontrolled hypertension, with systolic blood pressure ≥160 mmHg (but <180 mmHg) or diastolic blood pressure ≥100 mmHg (but <120 mmHg) lasting for ≥14 days or causing treatment delay for ≥4 days.
- Other Grade 3 non-hematologic AEs not related to underlying malignancy and lasting for >3 days despite optimal supportive care, with the following exceptions:

  - Fatigue that persists for ≤7 days;
  - Rash that resolves to Grade ≤1 within 3 weeks;
  - Elevation in serum amylase and/or lipase not associated with clinical or radiological evidence of pancreatitis;
  - Tumor flare (defined as local pain, irritation, or rash localized at sites of known or suspected tumor).

The following immune-related AEs were designated as DLTs:
- Grade 5 irAE;
- Grade 4 hematologic irAE, with the exception of:

  - Lymphopenia;
  - Neutropenia lasting ≤2 days that is not associated with fever or other clinically significant symptoms.
- Grade 4 non-hematologic irAE;
- Grade 3 non-hematologic irAE that does not resolve to Grade ≤1 or baseline with immunosuppressive therapy within 3 weeks, with the following exceptions:

  - Any endocrinopathy (thyroid, pituitary, and/or adrenal insufficiency) that is adequately controlled by hormonal replacement;
  - Tumor flare (defined as local pain, irritation, or rash localized at sites of known or suspected tumor);
  - Infusion-related reaction that resolves to Grade ≤2 within 6 hours;
  - Electrolyte imbalance/abnormality that is not associated with clinical sequelae and resolves within 3 days spontaneously or with supplementation/appropriate management.
- Grade 3 central nervous system (CNS)-related irAE regardless of duration or reversibility.
- Any Grade ≥2 irAE of eye pain or reduction of visual acuity that does not respond to topical therapy and does not improve to Grade 1 severity within 2 weeks of the initiation of topical therapy or requires systemic treatment.

Any other toxic effect during the DLT period may be assessed as a DLT if, upon review by the Investigators and Sponsor, it was agreed that the toxicity was of sufficient severity to be considered dose limiting.

The DLT evaluable population was defined as patients enrolled in any of the dose-escalation cohorts who experienced a DLT or who cleared the 9-week DLT period on triplet therapy without DLTs.

## Endpoints and assessments

The primary study objective was to evaluate safety of the triplet combination regimen, as characterized by type, incidence, severity, timing, seriousness, and relationship to study treatment of AEs, and laboratory abnormalities. The safety population was defined as all patients who received at least one dose of any study treatment drug (ie, sitravatinib, nivolumab, or ipilimumab). Secondary objectives and endpoints were clinical activity, as characterized by objective response rate (ORR), duration of response (DOR), disease control rate (DCR) termed in the trial protocol as clinical benefit rate (CBR) and defined as the percent of patients documented to have a best overall response of CR, PR, or SD; progression-free survival (PFS), one-year survival probability, and overall survival (OS); and pharmacokinetics (PK) of sitravatinib when administered in the triplet combination. Exploratory endpoints included gene expression signatures and immune cell populations in the tumour. Milestone 1-year survival probabilities were estimated using Greenwood's formula[40].

Imaging was used for disease assessments, with an allowable window of 4 weeks prior to first study treatment for screening/baseline evaluation, and first study scans at week 13 then every 8 weeks until week 49, after which disease assessments were performed every 16 weeks until objective disease progression. Baseline disease assessments were performed using computed tomography (CT), X-ray (radiography), or magnetic resonance imaging (MRI). Subsequent, on-study disease assessment included imaging of all known and suspected sites of disease identified in the preoperative setting (ie, CT or X-ray of the chest, CT, or MRI of the abdomen, and, if clinically indicated, whole body bone scan and CT with contrast or MRI of the brain and evaluation of any superficial lesions). Disease response was assessed per RECIST v1.1. Tumor measurements were performed by independent radiologists using the institutional Quantitative Imaging Analysis Core initiative established to provide standardized and reproducible imaging metrics across the University of Texas MDACC[41]. Blood samples for PK evaluation were collected at specified timepoints prior to and following study treatment dosing. Safety assessments were conducted at the initiation of study treatment and at each clinic visit. Per protocol, patients would be followed for survival every 2 months from the last study visit until death or loss to follow-up. Patients who were tolerating study drug regimen and deriving clinical benefit (based on at least one on-study disease scan) were eligible to participate on an open-label extension study at the time of 516-008 study closure to allow for continued access to sitravatinib and nivolumab.

## Clinical trial statistical analysis

The time-to-event Bayesian optimal interval (TITE-BOIN) design[29] was used to determine whether sequential dose escalation/de-escalation steps for sitravatinib in combination with nivolumab and ipilimumab described in Appendix 3 of the trial protocol (available in the supplementary note) should be undertaken and to identify the maximum tolerated dose (MTD). The TITE-BOIN design is well-suited for use in dose escalation studies involving treatments associated with late-onset toxicity, such as the triplet immunotherapy regimen evaluated in the present study, because it allows dose escalation decisions for new patients while some patients continue evaluation for DLT at the previous dose level, thus shortening the overall duration of the trial[29]. The model predicts the DLT outcome for ongoing patients based on their remaining follow-up time. Implementation is similar to the traditional 3 + 3 Phase 1 design but is more flexible and possesses superior operating characteristics comparable to more complex model-based designs[28,29,42].

The TITE-BOIN model implemented in this study was based on the following assumptions:
- The MTD is defined to have 0.3 probability of DLT;
- Initial cohort size is 3 patients;
- The overall duration of DLT assessment window is 9 weeks.

The approximate sample size was 27 DLT-evaluable patients, defined as patients enrolled in the Phase 1 dose-escalation portion of the study who experienced a DLT or who cleared the DLT period. Further information on the operating characteristics of the TITE-BOIN design for dose finding based on 1000 simulations of the trial, the decision schema and the decision table for escalation/de-escalation is included in the trial protocol available in the supplementary note.

## ScRNA-seq data processing and analysis

**Tissue samples.** We obtained tumor biopsies for scRNA-seq from 12 patients enrolled in the trial. Baseline samples were collected from all 12 patients, while 4 patients also had samples collected prior to the second infusion of nivolumab plus ipilimumab (C2 timepoint) and 3 patients also had samples at the end of treatment (EOT timepoint) due to disease progression. Whenever possible, longitudinally collected samples were biopsied from the same organ site. Supplementary Table S5 lists the timepoint and specific organ site biopsied for each patient sample.

**scRNA-seq library preparation and sequencing.** Tumor biopsy samples were thoroughly minced to size <1 mm and transferred to 2.0 ml Eppendorf tube containing 0.1 mg/ml Liberase TM and 0.2 mg/ml DNase I (MillieporeSigma) in RPMI1640/0.1% BSA. Tubes were then incubated at 37 degrees Celsius for 15 minutes with gentle rotation. Samples were filtered through 70 μm filters, spun down, and red blood cell lysis was performed with ACK lysis buffer (ThermoFisher, catalog #A1049201). Cell viability will be determined with Trypan blue exclusion assay. Only the samples with >60% viability were further processed on Chromium iX (10x Genomics) to generate 5' v2 gel beads-in-emulsions (GEMs). The barcoded cDNA was purified and amplified for the final generation of 10x single cell RNA sequencing (scRNAseq) libraries. After quantification with a high sensitivity DNA chip run on a Bioanalyzer 2100 system (Agilent), the libraries were sequenced as recommended by the manufacturer (approximately 50,000 reads per cell) on a NovaSeq 6000 system.

**Single-cell RNA data preprocessing and filtering.** The fastq files from single-cell sequencing were preprocessed using Cell Ranger software (version 7.1.0)[43] with default parameters. Reads were aligned to the human reference genome hg38 (version 2020-A, 10x Genomics) to calculate the expression matrix, which contained the number of unique molecular identifiers (UMIs) for every cell and gene. To ensure high-quality data for downstream analysis, careful selection of samples and cells was performed. Cells with gene expression of fewer than 500 genes were removed to eliminate cell debris, empty drops, and low-quality cells. Cells with a mitochondrial versus endogenous genes expression ratio exceeding 25% were also discarded. Additionally, genes expressed in fewer than 3 cells were excluded from the expression matrix. Subsequently, the Seurat package[44] was used for data preprocessing. The raw expression matrix was log2 transformed with the NormalizeData function, scaled with the ScaleData function using default parameters, and the top 2000 genes with the highest standardized variance were identified using the FindVariableFeatures function. Principal component analysis (PCA) was performed using the RunPCA function with default parameters.

**Batch correction.** Harmony[16] was applied with default parameters to correct batch effects within the PCA space during the major cell lineage clustering. Harmony was utilized for cell groups within the tumor

microenvironment to remove heterogeneity arising from different tissue origins. For epithelial cell populations, both pre- and post-Harmony correction results were retained to investigate tumor heterogeneity and explore potential commonalities in tumor progression. A careful evaluation of Harmony's clustering performance was conducted to ensure that different cell types were not overly corrected during the clustering process. The default parameters for Harmony during the batch correction of major cell populations effectively distinguished different cell types.

## Unsupervised clustering and differential gene expression (DEG) Analysis

RunPCA was performed on the top 2000 genes with the highest standardized variance, and the ElbowPlot function in Seurat was employed to generate an elbow plot. Based on this plot, the number of significant principal components (PCs) was determined. The FindNeighbors function in Seurat was utilized to construct a shared nearest neighbor (SNN) graph, derived from unsupervised clustering executed by the Seurat function FindClusters. Various resolution parameters for unsupervised clustering were examined, and cluster marker genes were reviewed to determine the optimal number of clusters representing distinct transcriptional profiles. For visualization purposes, dimensionality reduction was further achieved using the Uniform Manifold Approximation and Projection (UMAP) method[45], implemented through the Seurat function RunUMAP. Marker genes for specific clusters were identified using the FindAllMarkers function in the Seurat R package. DEGs were filtered with an expression fold change > 2 and FDR q-value < 0.01.

**Doublet detection and removal.** The removal of doublets or multiplets involved careful validation and screening. Some doublets or multiplets could form distinct cell clusters due to their unique heterozygous gene expression profiles, while others might be mixed into specific cell subpopulations. During major cell cluster classification, cells expressing markers in a specific cell type but showing elevated expression of markers for another cell type were selected. For instance, cells with high expression of epithelial-related markers in the T cell cluster and high expression of endothelial cell markers in the B cell cluster were chosen. Subsequently, the cell population was refined by excluding cells expressing different lineage-specific markers simultaneously. This step was repeated to ensure the exclusion of the majority of barcodes associated with cell doublets. Additionally, doublet detection tool DoubletFinder[46] was employed to predict potential doublets. The predicted doublets were thoroughly reviewed and compared with those defined by our criteria before their removal. After the removal of doublets, a total of 70,526 cells were retained for downstream analysis.

**Cell type identification.** Cell type identification was performed on the cell clusters identified after removing batch effects using Harmony. The identification process consisted of two steps. Firstly, identification of major cell groups, such as T cells, B cells, myeloid cells, and stromal cells, were identified. Subsequently, based on the first step, cell subtypes within each cell group were annotated based on their canonical markers. In each identification step, DEGs for each cell group were identified, and the top 30 most significant differentially expressed genes were reviewed. Meanwhile, feature plots and dot plots were visualized to demonstrate the expression characteristics of these genes in different cell groups. Cell groups were annotated based on the expression patterns of cluster-specific genes and well-known marker genes for different cell groups.

**Identification of copy number variants.** Single-cell copy number variants were inferred using inferCNV (version 1.16.0) R package. inferCNV estimates the genome copy number profile of single cells by exploring the expression intensity of genes across positions of the tumor genome in comparison to a set of reference 'normal' cells. Stromal cells were used as a reference to estimate the copy number alterations for epithelial cells. The reference annotation file of inferCNV was prepared following the guidance of its tutorial, and default parameters were used to identify the copy number variance.

**Quantification of group enrichment.** Utilizing the *Ro/e* methodology, where Ro/e denotes the ratio of observed to expected cell number[47], we assessed the enrichment or depletion of distinct cell subsets within specified groups. Succinctly, we determined the ratio of observed cell counts to random expectations through chi-squared testing across each cluster amidst diverse groups. An *Ro/e* value exceeding 1 signifies enrichment, whereas a value below 1 denotes depletion of cells within the designated group.

**Therapeutic index calculation.** The calculation method for the Therapeutic Index follows the description provided in previous literature[14]. The Therapeutic Index was formulated to evaluate if dynamic changes in cellular proportions correlate with clinical responses by assessing the relationship between cellular proportions and alterations in tumor size. Cellular proportion dynamics were determined by computing the relative fractions for immune cell clusters within corresponding major immune compartments. Changes in tumor size were evaluated by comparing relative changes between post- and pre-treatment samples. Specifically, the Therapeutic Index was defined as: Therapeutic Index = slope/$R^2$. Here, the slope and $R^2$ were obtained from a linear regression model ($lm(y2\sim x2)$) estimating the correlation between cellular proportions (x2) of corresponding immune cell clusters and changes in tumor size (y2) among patients treated with different regimens. A negative Therapeutic Index suggests that a higher fraction in cellular proportion for the corresponding immune cell cluster is associated with a more favorable clinical response, while a positive Therapeutic Index indicates that a higher fraction in cellular proportion is linked with a poorer clinical response.

**Signature score calculation.** Each major cell type comprises numerous subsets with different functional states. To assess the functional states of myeloid cell subsets, we collected a series of gene signatures, including M1, M2, angiogenesis, and phagocytosis-associated genes, from a 2021 study by Cheng et al.[48] For T cells, we obtained signatures representing different states such as naïve, TCR signaling, exhaustion, and cytotoxicity-related genes from a 2023 study by Chu et al.[11] The signature scores for these gene sets were calculated using the AddModuleScore function in the Seurat R package.

**Single-cell trajectories.** Pseudo-time trajectory analysis was performed with Monocle3 using default parameters[49]. In brief, the Seurat object was first converted to a Monocle CellDataSet. The raw count expression matrix was then pre-processed to normalize the data. Fifty PCs were used to capture validated genes across all the cells in the data set. After dimension reduction with the UMAP reduction method, the cells were put in order by how much progress they've made by the "learn_graph" function. Finally, the cells were ordered in pseudotime along a trajectory and visualized by the "plot_cell" function.

**Pathway enrichment analysis.** The Hallmark and Kyoto Encyclopedia of Genes and Genomes (KEGG) pathways downloaded from the Molecular Signature Database (MSigDB version 2023.2.Hs, https://www.gsea-msigdb.org/gsea/msigdb/)[50] were utilized for pathway enrichment analysis. Based on all differentially expressed genes obtained from the FindAllMarkers function for each cell group, we employed the enricher function from the clusterProfiler (version 4.8.3) package[51] to calculate the enrichment levels of these pathways. The

pvalueCutoff and minGSSize were both set to 1, while all other parameters were kept at their default values.

**Correlation of pathways with pseudotime.** To assess the relationship between pathway activity and pseudotime, we calculated the correlation coefficients for pathway activities and pseudotime. The pathway activity scores were calculated using the method mentioned above. For all cells, the Spearman correlation coefficient between pathway activity and pseudotime was determined using the cor function in R.

**Survival analysis in the TCGA KIRC Dataset.** The Log2-normalized gene expression matrix and overall survival information of the TCGA KIRC dataset were downloaded from the UCSC Xena browser (https://xenabrowser.net/datapages/). Utilizing the Malignant_C8 signatures obtained from the single cell data, we employed the ssgsea function of the GSVA package[52] to calculate the C8 signature score for each sample. Taking the C8 signature score as an example, patients were stratified into high and low-expression groups based on the median value of C8 signature score. Subsequently, we employed the log-rank test to calculate the p-value between the two groups in unadjusted analyses. We additionally generated Cox regression models adjusting for tumor stage and history of neoadjuvant therapy. The Kaplan-Meier method was utilized to visualize the survival curves.

### Reporting summary
Further information on research design is available in the Nature Portfolio Reporting Summary linked to this article.

## Data availability
The trial protocol is available with this submission as a Supplementary Note in the Supplementary Information file. All processed data are available at https://www.ncbi.nlm.nih.gov/geo/query/acc.cgi?acc=GSE264586 (GEO accession number GSE264586). Raw data are not provided due to deposited due to lack of patient consent. Requests to access data should be forwarded to the corresponding authors at PMSaouel@mdanderson.org and/or jgao1@mdanderson.org and/or lwang22@mdanderson.org. All requests for data and materials will be promptly reviewed to verify whether the request is subject to any intellectual property or confidentiality obligations. Any data and materials that can be shared will be released via a Material Transfer Agreement. Source data are provided with this paper.

## Code availability
All codes used to generate the scRNA-seq Figs. 3–6 and Supplementary Figs. S2-S9 are available for download at https://github.com/VividKai/TripleTrial_ccRCC.

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

## Acknowledgements

We thank all of the patients and their families who participated in this phase 1 study, Lisa Vuchak (Mirati Therapeutics, Inc.) for Medical Affairs support, Anish Vora (Mirati Therapeutics) and Kumar Ramaiyan (Mirati Therapeutics) for programming support. This study was sponsored by Mirati Therapeutics, Inc, now a subsidiary of Bristol Myers Squibb (BMS), and was supported by the National Cancer Institute through the Cancer Center Support Grant P30CA16672 (Institutional Tissue Bank (ITB) and Research Histology Core Laboratory (RHCL)). The study (Clinical-Trials.gov, NCT04518046) was designed by the academic authors and the sponsor Mirati Therapeutics, Inc., now a subsidiary of Bristol Myers Squibb (BMS). Data were collected by the academic authors and their research teams and were interpreted by the authors and the sponsor. The manuscript was written by the authors without medical writing support. The corresponding authors had full access to the data in the study and had final responsibility for the decision to submit for publication. The clinical and correlatives studies were additionally supported in part by MD Anderson's Prometheus informatics system and the Department of Genitourinary Medical Oncology's Eckstein and Alexander laboratories. Sequencing and data generation was supported by CA016672(ATGC) grant from The University of Texas MD Anderson Cancer Center, Advanced Technology Genomics Core. P Msaouel is supported by the National Cancer Institute R37CA288448, the Andrew Sabin Family Foundation Fellowship, Gateway for Cancer Research, a Translational Research Partnership Award (KC200096P1) by the United States Department of Defense, an Advanced Discovery Award by the Kidney Cancer Association, a Translational Research Award by the V Foundation, the MD Anderson Physician-Scientist Award, donations from the Renal Medullary Carcinoma Research Foundation in honor of Ryse Williams, as well as philanthropic donations by the Chris "CJ" Johnson Foundation, and by the family of Mike and Mary Allen. J Gao is supported by the Doris Duke Clinical Scientist Development Award (#2018097), the MD Anderson Physician Scientist Award, Khalifa Physician Scientist Award, Andrew Sabin Family Foundation Fellows Award, MD Anderson Faculty Scholar Award, the David H. Koch Center for Applied Research of Genitourinary Cancers, Wendy and Leslie Irvin Barnhart Fund, Joan and Herb Kelleher Charitable Foundation, KCA Advanced Discovery Award, the Williams TNT Fund, the V Foundation Translational Award, the DOD KCRP Translational Research Partnership Award and NIH/NCI R01 CA254988-01A1, R01 CA269489-01A1, and R01 CA282282.

## Author contributions

All authors collected and analyzed the data and provided critical review of the manuscript. P.M., C.D.C., and H.T. designed the clinical study, monitored, and interpreted clinical data. P.M., K.Y., J.G., and L.W. designed the translational correlative studies and interpreted their results. P.M., A.Y.S., A.J.Z., and T.N.M. enrolled patients on the study. P.M., K.Y., J.C., X.Y., M.K., F.D., R.A.S., P.R., K.S., M.L., C.Y., M.D., G.H., Y.C., M.H., P.O., D.S., and G.G. contributed to the collection, and analysis of the translational correlative studies. Y.Y. and R.Y., contributed to the statistical analysis. Development of the first draft of the manuscript was led by the corresponding authors. All authors contributed to drafting the manuscript and provided final approval. P.M., J.G., and L.W. had final responsibility for the decision to submit for publication.

## Competing interests

P. Msaouel has received honoraria for service on a Scientific Advisory Board for Mirati Therapeutics, Bristol Myers Squibb, and Exelixis; consulting for Axiom Healthcare Strategies and DAVA Oncology; non-branded educational programs supported by Exelixis and Pfizer; and

research funding for clinical trials from Takeda, Bristol Myers Squibb, Mirati Therapeutics, Gateway for Cancer Research, and the University of Texas MD Anderson Cancer Center. J. Gao serves as a consultant for AstraZeneca, Aveo Pharmaceuticals, CRISPR Therapeutics, Infinity Pharmaceuticals, Janssen, Jounce, Pfizer, Polaris, and Symphogen. N. M. Tannir reported receiving and personal fees (honoraria) from Calithera Biosciences during the conduct of the study; and grants (sponsored trial) from Calithera Biosciences, Bristol Myers Squibb (BMS), Nektar Therapeutics, Arrowhead Pharmaceuticals, and Novartis, as well as personal fees (honoraria) from Calithera Biosciences, BMS, Eisai Medical Research, Merck Sharp & Dohme (MSD), Deka Biosciences, Neoleukin Therapeutics, Exelixis, and Ono Pharmaceutical outside the submitted work. A.J. Zurita has served as an advisor for AstraZeneca, Bayer, Exelixis, Foundation Medicine, and Pfizer; reports payments for speaking or manuscript support from Amedco, CancerNet, Hikma, Janssen-Cilag, Mckesson Specialty Health, and Pfizer; and declares grants or contracts from Merck, Pfizer, Astellas, ABX, Clarity, and Curium outside the submitted work. M. Hallin, P. Olson, R.Yang, D. Slavin, H. Der-Torossian, and C. D. Chin are employees of Mirati Therapeutics, Inc. The remaining authors declare no competing interests.

## Additional information

[1]Department of Genitourinary Medical Oncology, The University of Texas MD Anderson Cancer Center, Houston, TX, USA. [2]Department of Translational Molecular Pathology, The University of Texas MD Anderson Cancer Center, Houston, TX, USA. [3]David H. Koch Center for Applied Research of Genitourinary Cancers, The University of Texas, MD Anderson Cancer Center, Houston, TX, USA. [4]Department of Genomic Medicine, The University of Texas MD Anderson Cancer Center, Houston, USA. [5]Department of Biostatistics, The University of Texas MD Anderson Cancer Center, Houston, TX, USA. [6]Department of Interventional Radiology, The University of Texas MD Anderson Cancer Center, Houston, TX, USA. [7]Department of Pathology, The University of Texas MD Anderson Cancer Center, Houston, TX, USA. [8]Mirati Therapeutics, Inc, San Diego, CA, USA. [9]The University of Texas MD Anderson Cancer Center UTHealth Houston Graduate School of Biomedical Sciences (GSBS), Houston, TX, USA. [10]These authors jointly supervised this work: Pavlos Msaouel, Jianjun Gao, Linghua Wang. ✉e-mail: PMsaouel@mdanderson.org; lwang22@mdanderson.org; jgao1@mdanderson.org

