## [Peer Review file · Nature Communications]

Sitravatinib in combination with nivolumab plus ipilimumab in patients with advanced clear cell renal cell carcinoma: a phase 1 trial

Corresponding Author: Dr Pavlos Msaouel

Version 0:

Reviewer comments:

Reviewer #1

(Remarks to the Author)

This is an important study regarding triplet therapy for mcrRCC using a backbone of ipi/nivo. The authors should be congratulated on completing an important study which is addressing a significant need in the field with this treatment, namely that the primary disease progression rate is high with ipi/nivo and the slow onset of action relative to other combination choices currently available.

I suggest that the authors add to the discussion the two most important limitations.

1- This is a highly toxic combination at the initial starting dose which required dose reduction of ipi. This possibly compromised the efficacy of the combination. Longer follow up will be needed to assess this potential risk to patients. The authors provide a sufficient rationale for why the dose modification was chosen and I agree with the rationale but still this limitation should be included.

2- Other prior TKI/IO combinations were similarly based on strong preclinical rationale that TKIs can induce a significant positive immuno-modulatory effect. However, none of these combinations appear to induce a clinically meaningful immunologic effect compared with PD-(L)1 therapy alone. It is possible that despite these measured immunologic changes that there will be no increased clinically observed immunologic activity compared to ipi/nivo alone. However, their data suggests that this particular TKI might be an exception since the irAE rate was elevated in contrast to all other TKI/IO doublets where the irAE rate is similar to PD-(L)1 monotherapy. However, this is still a limitation that should be mentioned.

Reviewer #2

(Remarks to the Author)

The authors have sufficiently revised the manuscript to address all reviewers' comments.

Reviewer #3

(Remarks to the Author)

The authors have responded to all my original concerns. I have no new concerns.

Point-by-point responses to reviewer comments:

We would like to thank the editor and reviewers for taking the time to thoughtfully review and give comments/suggestions to improve our manuscript. Please see our individual responses to the remaining reviewer comments below.

Reviewer #1:

1. This is a highly toxic combination at the initial starting dose which required dose reduction of ipi. This possibly compromised the efficacy of the combination. Longer follow up will be needed to assess this potential risk to patients. The authors provide a sufficient rationale for why the dose modification was chosen and I agree with the rationale but still this limitation should be included.

Authors: We have accordingly revised the Discussion section to note this limitation and that longer follow up will be needed to assess potential long-term risks to patients at all treated cohorts given the high immunogenicity of sitravatinib.

2. Other prior TKI/IO combinations were similarly based on strong preclinical rationale that TKIs can induce a significant positive immuno-modulatory effect. However, none of these combinations appear to induce a clinically meaningful immunologic effect compared with PD-(L)1 therapy alone. It is possible that despite these measured immunologic changes that there will be no increased clinically observed immunologic activity compared to ipi/nivo alone. However, their data suggests that this particular TKI might be an exception since the irAE rate was elevated in contrast to all other TKI/IO doublets where the irAE rate is similar to PD-(L)1 monotherapy. However, this is still a limitation that should be mentioned.

Authors: We have revised the Discussion section to note this limitation.